# A Qualitative Study of the Views of Health and Social Care Decision-Makers on the Role of Wellbeing in Resource Allocation Decisions in the UK

**Tessa Peasgood *** **, Jill Carlton** **and John Brazier**

School of Health and Related Research, University of Sheffield, Sheffield S1 4DA, UK;
j.carlton@sheffield.ac.uk (J.C.); j.e.brazier@sheffield.ac.uk (J.B.)

*  Correspondence: t.peasgood@sheffield.ac.uk; Tel.: +44-1142220677

**Abstract:** There has been growing international interest in the role that wellbeing measures could play within policy making in health and social care. This project explored the opinions of a sample of UK decision-makers on the relevance of wellbeing and subjective wellbeing (by which we mean good and bad feelings or overall evaluations of life, such as life satisfaction) for resource allocation decisions within health and social care. Through these discussions we draw out the perceived advantages and the potential concerns that decision-makers have about broadening out to wellbeing and subjective wellbeing rather than just measuring health. Three focus groups were conducted: with members of the National Institute for Health and Care Excellence (NICE) Citizen's Council, with a Health and Wellbeing Board at a Local Authority and with Public Health England. In addition, eleven semi-structured interviews were held with staff from NHS England and members of a range of NICE committees. We identified a range of opinions about the role of wellbeing and a broadly held view that there was a need for improved consideration of broader quality of life outcomes. We also identified considerable caution in relation to the use of subjective wellbeing.

**Keywords:** qualitative; subjective wellbeing; resource allocation; QALYs; health; happiness; UK; National Institute for Health and Care Excellence (NICE)

## 1. Introduction

Economic evaluation of healthcare commonly estimates the incremental cost per Quality Adjusted Life Year (QALY) of new health technologies. QALYs provide a way to capture both survival- and health-related quality of life (HRQoL) benefits. Whilst HRQoL may include all aspects of quality of life that could theoretically be impacted by health or health care, the term is often used to identify the subset of important ways in which health or health care impacts upon quality of life (Torrance 1987; Brazier et al. 2017). The rationale for taking a narrower perspective reflects the interests of healthcare policy-makers and clinicians and facilitates sensitive measurement of those aspects of quality of life most likely to change following treatment.

With greater policy emphasis on the coordination between health and social care, particularly for people with long-term conditions, there is a need to compare cost-effectiveness of interventions that span multiple sectors. A potential solution to cross-sector comparability is to rely on a broader quality of life or wellbeing measure (Brazier and Tsuchiya 2015). The terms wellbeing and quality of life are often used interchangeably as a broad judgement of how good an individual's life is. We can distinguish between a number of different theoretical conceptions of wellbeing (see Peasgood et al. 2014), but here we use the term wellbeing in its broadest sense without aligning to any particular theory of wellbeing.

There has been growing international interest in the role that wellbeing measures could play within policy making and health care in particular (Dolan 2008; Dolan and White 2007; Johnson et al. 2016; Lee et al. 2013; van de Wetering et al. 2016). In the UK this has been reflected in the Office for National Statistics (ONS) starting to monitor national wellbeing from 2011. The ONS gives particular emphasis to subjective wellbeing (by which we mean good and bad feelings or overall evaluations of life, such as life satisfaction), measured by four questions on life satisfaction, happiness yesterday, anxiety yesterday, and the extent to which the individual thinks they do worthwhile activities. The use of wellbeing measures to evaluate healthcare is not without its concerns (Mehta and Davies 2015; Hausman 2015). Policy-makers in health and social care wishing to assess any future role for wellbeing as an outcome measure face a number of uncertainties about the validity of wellbeing measures.

This project was commissioned by NICE to inform a dialogue around the use of wellbeing as an outcome measure. Qualitative work was conducted to explore the opinions of a sample of decision-makers on the relevance of wellbeing, particularly subjective wellbeing, for resource allocation decisions across different categories within health and social care. In this paper we present the findings of the interviews and focus groups undertaken about the relevance of wellbeing in resource allocation. We highlight the perceived advantages and concerns of relying upon subjective wellbeing, and give an indication of the degree of support across a sample of decision-makers for a move to a greater focus on wellbeing outcomes in preference to the narrower HRQoL.

## 2. Methods

Focus groups and interviews were conducted between January and March 2016 with stakeholders across health, public health and social care to explore their views on the most appropriate outcome measure for use in NICE health and social care guidance. This included discussions on the importance of subjective versus objective outcome measures. A provisional topic guide was developed amongst the research team. It was recognised that some of the participants may not be familiar with the concept of wellbeing and its measurement. Therefore, background material was developed to outline common ways of conceptualising wellbeing and to describe instruments currently used in the UK to measure wellbeing (background material available on request from the authors). Care was taken in the discussions to separate out the impact of interventions on longevity and on mortality, using examples where maximizing health or wellbeing would not impact upon longevity.

A pilot focus group was conducted with staff at the School of Health and Related Research (ScHARR) (many of whom had experience on NICE committees and working with policy-makers), to refine the background material and test the topic guide. These were shared with contacts at NICE (see topic guide in Appendix A).

Three focus groups were conducted: with members of NICE Citizen's Council (NICE CC) ($n = 7$), a Health and Wellbeing Board (HWWB) ($n = 10$) and Public Health England (PHE) ($n = 4$). In addition, 11 semistructured interviews (between 35 to 50 min) were held with staff from NHS England ($n = 3$) and members of NICE committees (NICE Com) ($n = 8$). The members of the NICE committees comprised Technology Appraisal Committees ($n = 2$), Highly Specialised Technologies Evaluation Committees ($n = 1$), Social Care Guideline Committees ($n = 2$) and Public Health Advisory Committees ($n = 3$). Participants had all responded to an internal email informing them of the study. Written consent was obtained from all participants and the study was approved by ScHARR ethics committee. All focus groups and interviews were recorded and transcribed verbatim.

The aim of the analysis was to understand the range of opinions on the relative importance of wellbeing, particularly subjective wellbeing relative to health outcomes, and to explore opinions on possible outcome measurement approaches. Thematic content analysis was undertaken using framework analysis (Spencer and Ritchie 2002). Framework analysis follows the principles of classifying data according to key themes, concepts and emergent categories. An initial framework was developed to guide the analysis. Two transcripts were examined independently by members of

the research team (TP and JC) to identify possible themes within the data. These themes were then compared between researchers and an initial thematic framework agreed. Each transcript was reviewed several times in order to become familiar with the data, and any new emergent themes were identified and discussed. The transcripts were then re-examined, and key phrases and sentences identified. All transcripts were independently coded according to the thematic framework (see Appendix B), compared and moderated.

## 3. Results

A number of themes emerged from the analysis around the definitions of wellbeing, conflicting opinions on the role of wellbeing within resource allocation decisions for public policy, concerns around the use of subjective wellbeing whilst simultaneously recognising its importance. These are presented and discussed below.

### 3.1. The Role of Health Versus Wellbeing Outcomes in Resource Allocation

There was a broad spread of opinion about the role of wellbeing, which mostly clustered into five views, with some participant's views fitting into more than one category:

View 1: Wellbeing should be the primary outcome of interest.

Some participants stated that measuring wellbeing was central for individual and population level resource allocation decisions, seeing it as the ultimate outcome of interest and arguing that it was important to look "at people as a whole" (NICE CC).

"the health service is there to enable people to play a meaningful role in society and to . . . . have a respectable level of sense esteem." (NICE Com: I.4)

"I think I'd rather use wellbeing across all areas than not at all" (NICE CC)

"Why do we do anything if it doesn't improve wellbeing, why improve health, why not have bad health?" (PHE focus group)

One participant raised the potential for wellbeing to act as a "common currency that all sectors—education, employment, health, environment—could use in order to evaluate the relative efficacy of their policies . . . If you had a measure of wellbeing that was applicable across all sectors . . . just think of the power of that" (PHE focus group).

View 2: Wellbeing is complementary information to health outcomes.

Some participants felt that wellbeing measures could help to give a fuller picture of the impact health conditions and interventions have on people's lives, and would therefore provide complementary information to health outcomes. One participant thought that within the current appraisal tools the "wellbeing and the person centeredness of care is the bit that is missing" (NHSE: I.1). Another noted that they would like to see "more discussion about wellbeing or a similar concept in NICE appraisals" (NICE Com: I.7).

Some saw a role for additional, more wellbeing focused items within core outcome measures, with "a role for wellbeing and for something like EQ-5D" (NICE Com: I.5).

Participants noted the bidirectional causal relationship between wellbeing and health. Enhancing wellbeing was seen as an important and under acknowledged means of health promotion, particularly in relation to behaviour change—

"wellbeing underpins all behaviour change you know you've got to feel that your life is worthwhile in order to stop smoking, for example". (PHE focus group)

View 3: Wellbeing is useful for resource allocation in some decisions but not others.

Those holding this view generally agreed that wellbeing was less relevant to evaluate interventions targeted at physical health or life extending interventions, where it played "a part, but a small part"

(NICE CC). But more relevant in areas such as palliative care, health conditions where health status could not be expected to improve, social care, looking at the impact on families, mental health (particularly depression) and in understanding the impact of the process of treatment.

Participants drew the distinction between interventions with a one-off health fix (such as a hip replacement) where health outcomes should be paramount and interventions for long-term conditions where the objective is management of the condition and supporting the patient's own goals in life where wellbeing outcomes would be more suitable (NHSE: I.1).

Interventions where no impact on health would be expected:

Many participants drew attention to situations where treatment and care is provided with no expectation of health outcomes changing, giving examples of dementia, post stroke, severe disability and end-of-life care. A reasonable aim in these circumstances was to make people "feel a lot better" (NICE CC). Some participants perceived the aims of social care as enhancing functioning and quality of life rather than changing health status hence using wellbeing as an outcome for social care provision "makes sense" (NICE CC). The ethos driving social care provision, which sees "the person as an individual with their own personal strengths and needs, their family and their community, and seeing them at the centre of that" (NICE Com: I.8) was seen as needing to prioritise wellbeing.

For end-of-life care the patient's own judgement of outcomes, and their "happiness perspective" (PHE focus group) was presented as the primary concern.

Interventions with an impact on families and carers:

Many participants felt the impact on families and carers was a relevant outcome for decision-making and recognised that the impact of caring "on health and wellbeing is very very substantial indeed" (NICE Com: I.3). Members of NICE CC showed recognition of the impact of health conditions (including mental health) and treatment was felt across the whole family and was in clear agreement that this family impact was relevant for decision-making. Another participant expressed that the relevant impact on carers extended beyond health to those aspects of life that the caring role "takes you away from" (NICE Com: I.2).

Interventions in mental health:

Participants noted that for many interventions in mental health, particularly where treatment plans are individualised, the aim is explicitly to improve quality of life, rather than improve any objective notion of health. Improving quality of life was stated as being the "whole point" (NICE CC) of interventions for depression.

Interventions with treatment burden:

Some participants discussed the relevance of the patient's time being taken up with treatment, particularly long hospital stays or frequent visits, noting the "impact of that isn't caught" (NICE Com: I.7), but that it could be included were a wellbeing approach adopted.

Interventions with a non health focus:

Participants often drew on their own experience with health conditions and health care interventions or those of friends and family to explain the importance of 'softer', non health-focused interventions (such as help with appearance, self-esteem, someone to talk to, general support, foot massages, provision of information and emotional support to prepare for treatment). Nonhealth interventions and provision for personal support were seen as particularly important for providing "ways to cope with things" (NICE CC). Even where interventions may not lead to a change in health status they were considered to deliver important outcomes. Some participants linked this to a placebo effect (NICE CC), or the value of simply feeling better regardless of clinical outcomes:

> "If there's no clinical evidence that there's been an improvement, but the person thinks through some mental process 'well I've been listened to, I've been talked to, I've reflected or whatever, and I feel better as a result of that' that is a positive outcome and that's something that has to be taken into account". (NICE Com: I.3)

View 4: Health outcomes should be the foundation of health care decisions

Those holding this fourth view thought that, although wellbeing was potentially interesting background information, health should still be "the foundation" of decisions (NICE CC), as it has a "primacy" where "getting cured is the most important thing" (NICE CC).

A couple of participants argued that whilst we can be certain that society values health and everyone will agree that more health is a good thing, we have little knowledge of society's views about public spending to enhance wellbeing (NICE Com: I.2), or whether using wellbeing instruments would result in "what the public thinks is a valid, worthwhile use of public money" (NHSE: I.2).

View 5: Wellbeing outcomes are not relevant due to a lack of reliable and valid evidence.

Participants holding this view felt that public policy decisions required peer-reviewed effectiveness evidence and unless the same robust evidence for wellbeing outcomes was available as it is for clinical and health outcomes it would not be possible to incorporate wellbeing. The QALY approach, using the EQ-5D, was seen as being fairly well understood and implementable, and no equivalent "equipment" (NICE Com: I.2) for anything beyond health was available. Validated instruments that capture quality of life across different conditions were viewed as unavailable.

One participant argued that whilst in the area of public health wellbeing was pretty important, there is an "absence of good scientific evidence" and no "vehicle for enabling coherent and robust consideration of wellbeing" (NICE Com: I:1). They also felt their current reliance on health outcomes arose in part "because the issues of wellbeing are just so incredibly tricky" (NICE Com: I.1).

*3.2. What is the Relevant Concept of Wellbeing for an Outcome Measure in Health and Social Care?*

Participants were probed to discuss the most relevant conception of wellbeing for health care appraisal, including their views about subjective wellbeing, particularly the use of happiness and life satisfactions questions. Participants described a number of different components of wellbeing that they perceived to be relevant to decision making including feeling happy and having fun (but not necessarily all the time), feeling content, coping financially, being able to do the things you want to do and find enjoyable, having the freedom to make decisions, feeling fulfilled with life, feeling that life has a purpose, having achievements, feeling valued, feeling like you have a role in life, autonomy, independence, being able to function (to sleep, eat, drink and move around) and not being in pain.

A key concept that was raised a number of times was that of coping: being able to cope and feel in control of your life, and having the "capacity to deal with life" (NICE Com: I.4), cope "with their condition" (NICE Com: I.7) and do what you want to do" (NHSE: I.1).

One participant defined wellbeing as "being able to maintain a state of physical, emotional and mental health" (NICE Com: I.6). Emotional health was seen as different to mental health and about general sense of how an individual was feeling (NHSE: I:3), tapping into something with a "lighter touch" than mental health—

> "you have to go quite a long way down the line to have damaged your mental health . . . whereas your emotional health is more about a kind of steady state and something about your relationships and so on". (NICE Com: I.6)

Participants grappled with the complexity of defining wellbeing and the tension between accepting individual differences in preferences versus universal human needs. Some emphasised the fact that people have different priorities and make different judgements in what is important in life—

> "it's very subjective as to what quality means to them, it could be spending time with their grandchildren, going on holiday, ticking a bucket list off" . . . "but for somebody who's like really close to God it could be spiritual wellbeing". (HWWB focus group)

However, some participants gave more credence to the notion that, broadly, as humans our needs and difficulties are the same—

> "The fundamentals that people worry about tend to be the same, am I going to be in pain, can I sleep, can I eat, can I drink, am I going to be lonely and then once you've established security

then we're up to can I get out of the house, can I go to social functions . . . I think actually the basic foundation really is pretty much the same for everybody" (HWWB focus group)

Personal relationships were an area in which the tension between identifying outcomes that are beneficial for all whilst at the same time appreciating people's differences was seen as being particularly acute. Engaging with others, strong personal relationships, having support from others and not being isolated or lonely were repeatedly raised as key outcomes. However, participants also showed awareness that some people choose not to have strong social connections, they may be very happy having little social contact or a small number of connections, or may choose not to have particular types of relationships (such as relationships with family members or intimate relationships). The important outcome was whether individuals had the kinds of relationships that they wanted—

"You might not want to talk to your father so it would need to be not that specific." (NICE Com: I.6)

The absence of "isolation" (NHSE: I.3) and opportunity to "connect", with the individual determining the nature of that connection (which may extend beyond people, e.g., to pets), was seen as a less prescriptive attribute.

A couple of participants raised the role of process outcomes such as personalised care and patient engagement as relevant considerations for assessment of the benefit of interventions for patients with long-term conditions. These process outcomes include aspects of dignity, respect, control over care and involvement in decision-making (NHSE: I.1). However, the extent to which these are inputs that lead to better clinical and quality of life outcomes or are outcomes in themselves was seen as conceptually difficult to determine (NHSE: I.1).

Regarding subjective wellbeing measures, some participants used the term 'subjective' in a problematic, critical sense and raised concerns about whether responses to subjective wellbeing questions, such as life satisfaction and happiness, could be compared between respondents. A number of participants used the term "nervous" (NICE Com: I.1, HWWB focus group) when referring to the idea of relying upon an individual's perceptions and judgements about their own lives, over more objective measures.

The effect of expectations on judgements about satisfaction with life and the potential for those with objectively poor health states to report satisfaction with life because of low expectations was frequently raised.

"So if your expectations are really very low, what does that do to your sense of wellbeing? Does it mean that because your expectations are low, and they haven't got any worse that you're feeling okay about that?" (NICE Com: I.3)

The potential biases and measurement error arising from questions asking about affect or emotion was frequently raised. These included the variability and day-to-day fluctuation of emotions, the potential lack of willingness of respondents to admit their lives were not good (NHSE: I.1) and the impact of non relevant or temporary contextual factors on people's responses (such as England football team winning a match (NICE Com: 1.3)). This was seen as particularly problematic given the necessity to sometimes deal with small sample sizes, where measurement error is more problematic than in a larger sample.

Participants also raised recall bias, particularly in relation to the ONS questions on happiness and anxiety yesterday. One participant noted that the recall to yesterday may generate systematic bias.

"it might just be a really rotten day yesterday and if you'd asked yesterday they'd have told you what a rotten day it was, but the halo effect is such that now that the bad things have gone out of my mind and nothing disastrous has happened so it was OK." (NICE Com: I.3)

One participant commented on the potential ambiguity in the ONS question on worthwhile activities—

"I would find it amazing that people had any sense of understanding of what that means ... I would imagine that people define that, if they define that at all, in very different ways" (NICE Com: I.3)

Consequently, interpreting the responses is problematic. Another participant gave a positive verdict on the ONS-4 question on worthwhile activities seeing it as able to allow for individual differences—

"I quite like that one because what I think is worthwhile is going to be very different from what you think is worthwhile" (NICE Com: I.6)

Objective measures, such as mobility and functioning, were seen as less subject to day-to-day fluctuation, less influenced by existing expectations, more interpersonally comparable and "easier to capture in a measure" (NICE Com: I.5). One participant felt focusing on basic needs had more legitimacy than happiness noting "I'm not going near your spirituals" (HWWB focus group).

One participant felt that more objective aspects of wellbeing and opportunity were more relevant outcome measures for health and social provision, noting that happiness was—

"a personal subjective thing and it's not necessarily found by virtue of what the state can provide ... So, you know, in asking about happiness ... you've got I suppose a risk of setting out to be the provider of the means by which people can find that and I don't necessarily think that's what the ... state system really should necessarily be doing, but at the same time we should be providing, kind of attending to the needs of people who are significantly disadvantaged ... so that they can function on the same level as the rest of the community ... and from that might come happiness" (NICE Com: I.8)

Another participant felt that a move to rely upon happiness as an outcome measure would for many decision-makers in health care be "one step too far" (NHSE I.1).

*3.3. Views on the Ways Forward for UK Resource Allocation Decisions within NICE*

While appreciating the divergence of opinion of what constitutes wellbeing, some participants argued that it was unmanageable to incorporate wellbeing into decision making without adopting a clear central position on what wellbeing is and a clear definition of wellbeing. Some argued that this conceptual clarity should be driven from the top, with NICE adopting a clear theoretical position supported by normative reasoning. Others saw a role for the public in establishing "the kind of wellbeing they want to encourage and promote" (PHE focus group).

Participants did not perceive that an ideal measure of wellbeing or "silver bullet" (PHE focus group) would be found. A pragmatic approach to providing outcome measures with the potential to support health and social care interventions and support evaluation of personal budgets was seen as the way forward.

"At the moment there's just so many different tools out there, I think it's crying out for something to be done ... something that enables some consistency and coherence" (NICE Com: I.1)

Participants raised concerns on the inconsistency between the wellbeing information presented to NICE committees, which often comes from a small number (1 to 3) of patient representatives, versus validated health data drawn from clinical research. They felt that patient representatives, who are typically more articulate and potentially had better outcomes with the intervention in question, may not be representative of the typical quality of life impact of the intervention; yet representative quality of life data for patients was not available. This was presented as a frustrating situation for both patient representatives, who may feel their judgement on their own life is not being taken into consideration, and committee members, who are not presented with the full evidence necessary to make good decisions—

"there is quite a big discrepancy often between the verbal evidence that is given by a patient expert . . . and the wellbeing data that goes into eventual decision making. This concerns me because I think that . . . it doesn't seem to be taken into account in the actual process very well". (NICE Com: I.7)

One participant noted that the impact on carers often gets overlooked at NICE committee level due to an absence of evidence—

"because we don't put a number on it, it just gets lost". (NICE Com: I.5)

One participant noted that providing wellbeing data just for background information is easy to dismiss, hence collecting this data under these circumstances would be an inefficient use of funds (NHSE: I.3). Only if wellbeing can be measured in a meaningful, comparable way will it be useful to guide resource allocation. (NHSE: 1.3)

Participants raised a number of requirements that they would wish for in an outcome measure. These were reflective of best practice in outcomes measurement, and included that an instrument be: comprehensive, valid, interpersonally comparable, sensitive to change, reliable and practical (short and simple). Some participants argued that sensitivity to change should include, not just improvements in underlying 'within skin' abilities, but differences in functioning that could be due to changes in the amount or type of support provided (such as having technical or personal support to be able to get out to the shops).

Participants frequently referred to the need for instruments with evidence to show their validity for use in particular areas and patient groups (although no participants elaborated on the type of validity evidence they would like to see).

## 4. Discussion

This research identified a wide range of opinions about the use of wellbeing in health care resource allocation ranging from substantial support to substantial reluctance. Participants in the PHE focus group raised the general divide between public health and social care practitioners who adopt a more pro-wellbeing approach and the current incentives within the healthcare system that are driven by a more medicalised, health-focused model. They did, however, also note the change within the UK over the last few decades in relation to the rising importance of "patient experience" (PHE focus group). They also noted a divergence of views on how wellbeing should be measured and the role of subjective wellbeing within public health, noting the lack of a "shared view in the sector" (PHE focus group).

Overall, participants working within public health, mental health and social care tended to adopt a more favourable attitude towards the use of wellbeing to evaluate interventions, although there was still divergence within these groups. Most participants included in this study did not feel wellbeing outcomes were being adequately captured within current resource allocation decisions. The lack of balance between patient representation, which provides information about quality of life and wellbeing impacts without the use of a valid outcome measure, and the relative strength of evidence presented in terms of the health-focused cost/QALY has been identified in other qualitative work exploring patient roles on NICE committees, with direct patient experience being seen as "peripheral, perhaps even tokenistic" (Hashem et al. 2018).

It should be noted that this sample is not necessarily representative of views of decision-makers across health and social care in the UK. Whilst an attempt was made to encourage those with a range of opinions to participate, those with a stronger current interest in wellbeing may have had more of an incentive to participate. Despite this possible selection bias there was still little appetite for an outcome measure solely focused on subjective wellbeing. Whilst a desire for broader outcomes than just health was clearly present, subjective wellbeing was only seen as part of the appropriate outcomes, which included physical and mental health and social and emotional wellbeing. Participants discussed the challenge of respecting the uniqueness of what matters to individuals whilst maintaining

interpersonally comparable outcomes. The concerns raised here around relying on subjective measures, and the use of life satisfaction in particular, are reflective of the broader academic debate on the use and validity of these measures and appropriateness of their underlying theories (see Haybron 2016 for a summary).

One of the initial intentions for the interviews was to gather in-depth views on the particular conception of wellbeing that participants considered important. However, discussions around these theoretical distinctions were not very fruitful. For example, many participants considered physical functioning to be an important outcome but did not clearly distinguish between whether it was important because physical functioning impacts upon how individuals think and feel about their current life (and/or future life), or whether they are important in an intrinsic, noninstrumental sense.

Although the intention within the discussion was to focus on exploring the content of the Q part of the QALY (should it be health, health-related quality of life, broader quality of life/wellbeing or subjective wellbeing?), and examples were explicitly chosen where years of life were held constant, the link between improving health and extending years of life makes it difficult to know for certain whether comments about the "primacy" of health for example are being driven by the importance of extending years of life.

The comment from the NICE Citizen's Council focus group about the importance of getting "cured" lends some credence to the concerns that the public may not support the legitimacy of public funds being used to enhance wellbeing over health. The example of cancer treatment was frequently referred to where getting cured, or improvements in clinical health outcomes, was seen as more closely aligning to additional years of life, whereas wellbeing outcomes were seen as important but softer, and less associated with additional years of life. This again emphasises the difficulty of isolating views on just the Q aspect of the QALY. In addition, minimal discussion was had regarding whether any preference for health outcomes over broader wellbeing outcomes was linked to equity rather than efficiency concerns.

The discussions held here deliberately set out to use the term 'wellbeing' in a broad sense to allow an exploration of ideas, remain open to alternative conceptions and allow the common usage of the term. However, this comes at a cost of potential lack of clarity. This is a more universal issue, with considerably mixed, overlapping and unclear usage of the terms 'wellbeing', 'subjective wellbeing', 'quality of life', 'health-related quality of life' and 'health status' both amongst health care researchers and decision-makers (Cummins et al. 2004; Salvador-Carulla et al. 2014; Vermeulen and Krabbe 2018; Aidem 2017; Karimi and Brazier 2016) and at a policy level (e.g., NHS England 2014). When discussing at the measurement level, such as the use of a question about happiness or the five-dimension EQ-5D, there is far greater clarity about what is, and what is not, included in health and wellbeing. At the measurement level clearer theoretical and empirical differences between these concepts emerge (Dolan and Metcalfe 2012; Dolan et al. 2017; Mukuria et al. 2016). Yet without the explicit link to specific measures the terms could be referring to the same or quite different things. A survey among a diverse group of 140 healthcare decision-makers from 23 countries identified a core set of shared decision criteria in which clinical efficacy and effectiveness were the main criteria (Tanios et al. 2013). How broad the concept is that is incorporated within the perception of 'effectiveness' clearly matters. There may be some areas that would not be expected to be covered within standard measures of effectiveness, such as patient convenience—for example, in a review of the literature on the relevant criteria for decision making among stakeholders convenience and dignity were identified as additional criteria separate to health status (Vermeulen and Krabbe 2018). But others, such as aspects of positive emotional experience or impact upon relationships, may or may not be seen as relevant depending on how 'effectiveness' is interpreted. The discussions held within this study identified need for greater breadth of outcome over a narrow conception of health status described using EQ-5D, although support for wellbeing or subjective wellbeing outcomes was far greater in some conditions/areas than others.

The mixed role for wellbeing across different sectors and conditions raises a tension between consistency between different areas and appropriateness of the outcome measures within each area.

It should be noted that the aim here was just to present a snap shot of a sample of decision-makers opinions and no attempt was made to debate or challenge respondent's views or ask them to explore trade-offs between sector appropriateness and broader consistency objectives.

## 5. Conclusions

This study explored the views of a sample of decision-makers across health and social care in the UK, focusing on their opinions on the appropriate outcome measure that NICE should use to support resource allocation decisions.

We identified a broadly held view that there was a need for improved consideration of broader quality of life outcomes than used at present. This was particularly apparent in areas where health improvement is not the key objective of interventions. We also identified considerable caution in relation to the use of subjective wellbeing and a reluctance to rely only on self-reported happiness or life satisfaction due to concerns over interpersonal comparability. Similar work is needed with key decision-makers in other countries to corroborate these findings.

**Author Contributions:** Conceptualization, T.P., J.C. and J.B.; Formal analysis, T.P. and J.C.; Funding acquisition, J.B.; Methodology, T.P., J.C. and J.B.; Writing—original draft, T.P.; Writing—review & editing, T.P., J.C. and J.B.

**Funding:** This research was funded by the National Institute for Health and Care Excellence (NICE) through its Decision Support Unit grant number [146789]. The views, and any errors or omissions, expressed in this document are of the authors only. NICE may take account of part or all of this document if it considers it appropriate, but it is not bound to do so.

**Acknowledgments:** We would like to thank the participants who kindly gave up their time and contributed to the focus groups and interviews. We would also like to thank Beth Shaw, Gill Fairclough, Milad Karimi, Clara Mukuria and Liam Wright for their helpful comments.

**Conflicts of Interest:** The authors declare no conflict of interest.

## Appendix A. Topic Guide

Are wellbeing outcomes currently used in resource allocation decisions you have been involved in?
Should we take account of wellbeing in deciding to fund: [probe]

1. A new chemotherapy drug for breast cancer?
2. A new treatment for depression?
3. A new enhanced follow-up service for stroke?

- Only use impact on wellbeing
- Use impact on wellbeing and health
- Do not use wellbeing (only health)

What sort of wellbeing should be considered?

a. evaluations (e.g., how satisfied are you with your life)
b. feelings (e.g., how happy are you, how worried you are) [experienced in the moment]
c. flourishing (e.g., psychological functioning such as feeling in control or feeling loved)
d. capability (e.g., the opportunity to have relationships/to have security)
e. objective list (e.g., having friends, time spent doing activities)

What do you think about the ONS-4 questions?
If familiar with EQ-5D, or WEMWBS what do you think about them?
What about impact on family and carers?
Do you think it might be different for social care and public health?
What would your view be of interventions in which objective health outcomes showed no improvement but subjective wellbeing outcomes showed improvement?
Present scenario:

- Intervention A versus B
- Long-term treatment outcomes and overall costs identical
- A required a longer time spent in hospital?
- Which would be preferred and why? [probe]

**Appendix B. Themes Extracted from the Focus Groups and Interviews**

1. Role of wellbeing in health and social care resource allocation

    1a. Importance in situations where health cannot change (e.g., dementia and long-term conditions)
    1b. Importance of wellbeing in social care
    1c. Importance of wellbeing to pick up impact on carers/family members
    1d. Importance in depression/mental health
    1d. Importance to the patient of interventions designed predominantly to make people feel good (e.g., placebo, foot massage, someone to talk to, etc.)
    1e. Variability of role of wellbeing depending upon context
    1f. Wellbeing data providing a fuller picture/wellbeing is important for resource allocation
    1g. Continue to require health information
    1h. Health is the overriding concern
    1i. Wellbeing data useful at population level
    1j. Wellbeing impact of the process of treatment
    1k. Wellbeing evidence is not available for NICE committees/or other current resource allocation decisions

2. The relationship between health and wellbeing

    2a. Wellbeing as a causal mechanism to improved health—or cost savings
    2b. Health and wellbeing as nonseparable or separable
    2c. Health is causal to wellbeing

3. Requirements of a wellbeing instrument

    3a. Addressing the fact that wellbeing is different [or the same] for different people
    3b. Valid

    3.b.1 Does the question mean to people what we think it means?
    3.b.2 Call for evidence on validity

    3c. Robust (e.g., use a number of questions)/use extra questions as cross-reference
    3c. Single score
    3d. Keep it not too complicated
    3e. Information to help patients see quality of wellness during and after treatment to support patient decisions
    3f. Comparability
    3g. Able to show improvement
    3h. More items—enough detail for understanding
    3i. Linking to provision and quality of care provided
    3j. Contains both objective and subjective
    3k. Goes beyond health—to other budget commitments of local government/addressing social determinants of health

4. Components of WB instrument

4a. Personal relationships/isolation/loneliness/engaging with people/support

4b. Happiness (even if not all the time)/content/fun

4c. Feeling fulfilled/doing worthwhile activities

4d. Freedoms

4e. Financial

4f. Control/coping

4g. Social activity

4h. Doing things you want to do

4i. Health and functioning (pain, sleep, eating and drinking)/Maslow's hierarchy of needs

4j. Self-esteem/goals and motivation/feeling good about themselves

5. Concerns with subjective wellbeing (life satisfaction/happiness/ONS questions)

5a. Role of expectations

5b. Influence of temporary/contextual factors to responses/influence of biases/wide interpretation (and counter argument)

5c. Recall concerns for yesterday and biases introduced by recall

5d. Not knowing what society thinks about wellbeing

5e. Positive judgement on ONS questions

5f. Too subjective—non-interpersonally comparable

5g. Unclear, nonintelligible, lacks face validity, ambiguous

6. Ways forward to incorporate wellbeing

6a. Role of Citizen's Council/Juries

6b. Normative stance needs to be adopted—including defining wellbeing

6c. Pragmatic steps (e.g., potential for evaluating personal budgets, need to integrate social care, etc.)

6d. Understand trade-offs of improving wellbeing versus extending life

6e. Pragmatic measurement approach necessary

6f. Concern with role of patient representatives in NICE committees

6g. Need for evidence

7. Issues relating to EQ-5D or WEMWBS

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
