# Peer review of "A Qualitative Study of the Views of Health and Social Care Decision-Makers on the Role of Wellbeing in Resource Allocation Decisions in the UK"

_economies, doi:10.3390/economies7010014_

Round 1
Reviewer 1 Report
Economies-373558: A qualitative study of the views of health and social care decision makers on the role of wellbeing in resource allocation decisions in the UK
The paper explicitly takes up the role of wellbeing in resource allocation among various relevant decision makers. The background is nicely put forth to advance the information on the topic but what I see lacking is the definition of wellbeing as to what actually wellbeing is and how it is related to any decision making process in any sector of an economy. Although authors do state the nature of wellbeing concept adopted for the study but still there lacks a clarity in its relevance to the pertinent topic. Same is observed in sub-section 3.1 where just wellbeing outcomes are taken just as granted. Similarly, there is some redundancy at few sections which also needs to be fixed. In my view, sub-section 3.3 can be integrated with the preceding one by shortening both to conserve space and idea. As in discussion, there is only one reference but one would expect some more insights from previous research to validate the discussion and for easy referencing. In the end, in my view, at many places, authors create a confusion by mixing health outcomes and wellbeing as for example in the later part of discussion. There is a typo in conclusion ‘use to use to support…’.
Author Response
The paper explicitly takes up the role of wellbeing in resource allocation among various relevant decision makers. The background is nicely put forth to advance the information on the topic but what I see lacking is the definition of wellbeing as to what actually wellbeing is and how it is related to any decision making process in any sector of an economy. Although authors do state the nature of wellbeing concept adopted for the study but still there lacks a clarity in its relevance to the pertinent topic. Same is observed in sub-section 3.1 where just wellbeing outcomes are taken just as granted.
This is an interesting point, and one that was discussed at length throughout the project.
We noted in the earlier submitted draft that because of the potential lack of familiarity with the term ‘wellbeing’ or ‘subjective wellbeing’ we gave an introductory presentation to focus group and on-line to interveiw respondents which briefly set out the meaning of these terms and different theoretical constructs (hedonism, objective list, capabilities, psychological wellbeing, subjective wellbeing, desire satisfaction). This presentation also described the Office for National Statistics approach to measuring wellbeing and other key relevant measures to provide useful background for discussions. For further clarity we have now included a copy of this presentation as an appendix.
We have also added an additional paragraph within the discussion which addresses this issue and places this limitation within the broader need for greater clarity across the board in how these terms are applied. Indeed, greater definitional clarity from above (e.g. NICE) was a call made by one of the participants.
Similarly, there is some redundancy at few sections which also needs to be fixed. In my view, sub-section 3.3 can be integrated with the preceding one by shortening both to conserve space and idea.
Thank you - this has been done.
As in discussion, there is only one reference but one would expect some more insights from previous research to validate the discussion and for easy referencing.
We have included some more comparisons to previous research and relevant references within the decision section. This includes reference to other qualitative work exploring the views of decision makers relating to resource allocation within healthcare - both in the UK and internationally.
However, it is worth noting that we were unable to identify much directly comparable research that discusses the role of wellbeing and the relevant concept of wellbeing for healthcare resource allocation.
We have also now included additional references elsewhere within the paper.
In the end, in my view, at many places, authors create a confusion by mixing health outcomes and wellbeing as for example in the later part of discussion.
We have aimed to address this through an additional discussion. Whilst we do not dispute that the confusion is present we think it is inherent in the historic and current ways in which this terminology is used. We think it is important to document
- divergent ways in which healthcare decision makers interpret wellbeing
- divergent opinions on the relevance of subjective wellbeing or broader wellbeing conceptions for decision making.
There is a typo in conclusion ‘use to use to support…’.
Thank you - this has been changed.
Reviewer 2 Report
The paper, to be published, needs to be more appropriately structured. First of all, in the introduction should refer to the objectives of the paper, which must be clearly outlined, rather than referring to a project. The methodological and results sections are well developed. The discussion is affected by the lack of a clear identification of the research objectives. Therefore, it must be enriched by comparing the results achieved with the available literature.
Author Response
The paper, to be published, needs to be more appropriately structured. First of all, in the introduction should refer to the objectives of the paper, which must be clearly outlined, rather than referring to a project.
This has been done. The introduction now states
“In this paper we present the findings of the interviews and focus groups undertaken. We highlight the perceived advantages and concerns of relying upon subjective wellbeing, and give an indication of the degree of support across a sample of decision makers for a move to a greater focus on wellbeing outcomes in preference to the narrower HRQoL.”
The methodological and results sections are well developed. The discussion is affected by the lack of a clear identification of the research objectives. Therefore, it must be enriched by comparing the results achieved with the available literature.
We have now added additional reference to comparable literature within the discussion section. However, as noted in the response to Reviewer 1 we were surprised by how little directly comparable research we were able to find, where the relevance of wellbeing for resource allocation was the main aim of the study.
Round 2
Reviewer 1 Report
Authors have nicely addressed my points and now I recommend the article for further processing by the journal.
Reviewer 2 Report
Thank you for providing a revised version of your paper that, now, is publishable. I would suggest to remove the term "project" (in the abstract and line 58) and refer, instead, to a paper that is based on a project. Good luck!